# Customized Subperiosteal Implants for the Rehabilitation of Atrophic Jaws: A Consensus Report and Literature Review

**DOI:** 10.3390/biomimetics9010061

**Published:** 2024-01-22

**Authors:** Javier Herce-López, Mariano del Canto Pingarrón, Álvaro Tofé-Povedano, Laura García-Arana, Marc Espino-Segura-Illa, Ramón Sieira-Gil, Carlos Rodado-Alonso, Alba Sánchez-Torres, Rui Figueiredo

**Affiliations:** 1Oral and Maxillofacial Surgeon, Virgen Macarena University Hospital, 41009 Seville, Spain; doctorherce@gmail.com; 2Private Practice Clínica del Canto, 28290 Madrid, Spain; mcantop@telefonica.net; 3Oral and Maxillofacial Surgeon, Puerta del Mar University Hospital, 11009 Cádiz, Spain; alvarotofe@gmail.com; 4Oral and Maxillofacial Surgeon, San Francisco de Asís University Hospital, 28002 Madrid, Spain; garcia.arana@imaxde.es; 5Oral and Maxillofacial Surgeon, Bellvitge University Hospital, 08907 Barcelona, Spain; marcespinosegurailla@gmail.com; 6Oral and Maxillofacial Consultant, Hospital Clínic, Universitat de Barcelona, 08036 Barcelona, Spain; ramonsieiragil@me.com; 7Oral and Maxillofacial Surgeon, Private Practice Cimax, 17002 Girona, Spain; crodadoa@gmail.com; 8Professors of Oral Surgery, Faculty of Medicine and Health Sciences, Researchers at the IDIBELL Institute, University of Barcelona (Spain), 08907 Barcelona, Spain; ruipfigueiredo@hotmail.com

**Keywords:** dental implants, customized subperiosteal implants, edentulous jaw

## Abstract

(1) Background: The aim was to perform a literature review on customized subperiosteal implants (CSIs) and provide clinical guidelines based on the results of an expert consensus meeting held in 2023. (2) Methods: A literature search was performed in Pubmed (MEDLINE) in July 2023, including case series and cohort studies with a minimum follow-up of 6 months that analyzed totally or partially edentulous patients treated with CSIs. Previously, an expert consensus meeting had been held in May 2023 to establish the most relevant clinical guidelines. (3) Results: Six papers (four case series and two retrospective cohort studies) were finally included in the review. Biological and mechanical complication rates ranged from 5.7% to 43.8% and from 6.3% to 20%, respectively. Thorough digital planning to ensure the passive fit of the CSI is mandatory to avoid implant failure. (4) Conclusions: CSIs are a promising treatment option for rehabilitating edentulous patients with atrophic jaws; they seem to have an excellent short-term survival rate, a low incidence of major complications, and less morbidity in comparison with complex bone grafting procedures. As the available data on the use of CSIs are very scarce, it is not possible to establish clinical recommendations based on scientific evidence.

## 1. Introduction

Dental implants are one of the main options for rehabilitating totally edentulous patients. However, in cases of severely atrophic maxillae or mandibles, the available bone might be insufficient for the placement of these medical devices. In these situations, bone grafting procedures might be indicated. Nevertheless, these techniques can be complex and usually require a longer treatment time [1]. When upper arches are involved, zygomatic implants can be used since they have good clinical outcomes and allow immediate loading [2]. However, it is important to stress that zygomatic implants have also been associated with several complications, some of which can be quite difficult to manage [3].

The development of new technologies has made it possible to manufacture customized implants to rehabilitate patients in whom standard implants cannot be placed because of trauma, oncological treatments, or malformations. These customized subperiosteal implants (CSIs) are designed for the patient’s specific anatomy and enable the selection of the most suitable anchorage areas. Furthermore, these structures facilitate rehabilitation since the professional can choose the position and type of prosthetic connection, allowing optimal force distribution [4,5,6,7,8]. Indeed, CSIs can support fixed prostheses with similar characteristics to those fabricated over conventional dental implants, even using an immediate loading protocol [4,5,6,7,8,9,10,11,12]. Moreover, the survival rate of CSIs seems to be high, and the most common complication described is exposure of the structure due to soft tissue dehiscence [1,12,13,14].

Since CSIs are a recent development, the literature on this topic is still quite scarce. Thus, the aims of this paper were to perform a literature review on CSIs and to provide clinical guidelines based on the results of an expert consensus meeting held in 2023.

## 2. Materials and Methods

A literature search was performed in Pubmed (MEDLINE) in July 2023 using the following search strategy: “customized subperiosteal implant OR subperiosteal personalized implants”. All case series and cohort studies with a minimum follow-up of 6 months that analyzed totally or partially edentulous patients treated with CSIs were included. Case reports and animal studies were excluded. The level of evidence from the selected studies was assessed using the SIGN guidelines [15].

A group of experts was selected to discuss the main aspects related to the use of CSIs to rehabilitate atrophic jaws. The workgroup included experienced professionals in the areas of oral and maxillofacial surgery, prosthodontics, dentistry, research methodology, and biomedical engineering. Initially, the clinicians involved were asked to analyze the most relevant papers on this topic. An on-site consensus meeting was then held in May 2023 in Santpedor (Manresa, Spain) to discuss the most relevant aspects in the following areas of interest:Indications and contraindications of CSIs;Planning and designing CSIs;Surgical protocol and associated complications;Prosthetic protocol and associated complications;Peri-implant supportive therapy;General recommendations and future perspectives.

All the participants had the opportunity to share their clinical experiences during the meeting. Furthermore, several cases were presented and examined by the clinicians involved, focusing especially on the above-mentioned areas of interest. If the participants had different opinions on a specific topic, a consensus was reached. A document with the main recommendations and conclusions was then prepared and sent to all authors for review. Afterward, a second online meeting was held in September 2023 to analyze all the recommendations and discuss the issues raised during the review process. A final report was prepared and sent to all the authors for their final approval.

## 3. Results

### 3.1. Literature Review

The electronic search yielded a total of 327 references. After duplicate removal and assessment of the titles and abstracts, 14 papers in total were selected for full-text analysis. Six papers [12,16,17,18,19,20]—four case series and two retrospective cohort studies—were included in the review (Table 1). The number of patients treated ranged from 4 to 70.

Five of these six papers analyzed laser sintering titanium CSIs [12,17,18,19,20], while in Rams et al.’s study [16], the implant frames were cast from a cobalt–chromium–molybdenum alloy.

The most frequent indication for using CSIs was the rehabilitation of full mandibular and/or maxillary edentulous patients. Other indications were also mentioned, like the treatment of severe defects after oncological surgical treatments and patients unwilling to undergo complex regenerative procedures.

The papers did not mention significant intraoperative complications, although some referred to discomfort and swelling in the early postoperative period (initial 2 weeks). Postoperative soft tissue dehiscence was a common finding. In this regard, Nemtoi et al. [19] and Dimitroulis et al. [20] reported 37.5% and 23.8% CSI exposure rates, respectively. Biological complications, including soft tissue dehiscence, peri-implantitis, and implant failure, varied from 5.7% [12] to 43.8% [19]. Mechanical complications, frequently related to the provisional prosthesis, ranged from 6.3% [19] to 20% [16]. Nevertheless, it is important to stress that most of the studies had a follow-up period of 2 years or less.

### 3.2. Clinical Guidelines Based on the Results of the Consensus Meeting

As mentioned in the Materials and Methods, the experts wrote a document based on the available literature [12,16,17,18,19,20]. If published data were lacking or considered insufficient, expert opinions (EO) were given. The following recommendations were made for the use of CSIs to treat edentulous patients:
1.Indications and contraindications of CSIs.
1.1.CSIs indications:
1.1.1.Patients who present insufficient bone to place standard dental implants;1.1.2.When complex regenerative techniques cannot be performed or are not accepted by the patients because of the associated morbidity;1.1.3.Patients who do not tolerate removable prostheses or when these cannot be made;1.1.4.CSIs might be considered as an alternative to zygomatic implants when a fixed prosthesis is required;1.1.5.CSIs should be used with caution in cases of partial edentulism since the available clinical data are limited in these situations (EO).1.2.CSI contraindications:
1.2.1.Patients with systemic pathologies that contraindicate the surgical procedure;1.2.2.Patients under treatment with therapies or drugs that contraindicate the surgical procedure.2.Planning and designing CSIs.
2.1.A thorough diagnosis is paramount for adequate treatment planning. High-resolution computer tomography (CT) following the instructions provided by the CSI manufacturer is mandatory. Cone-beam computer tomography (CBCT) is not suitable for designing CSIs (EO);2.2.A proper diagnosis should include the occlusal position, a standard tessellation language (STL) file with the intraoral anatomy, and a CT scan;2.3.Passive fit of the CSI to the surrounding bone is critical since this is a custom-made device;2.4.Since the most frequent complication is CSI exposure, a polished titanium surface is recommended (EO);2.5.It is essential to avoid abrupt transitions and sharp angles in the areas between the CSI frame and the prosthetic connections (Figure 1);2.6.Fixation of the CSI is a key factor for achieving a successful treatment outcome. The fixation elements should be placed in high anatomic buttress areas (nasal and zygomatic) and the palatal region. The use of self-drilling screws is recommended;2.7.In cases with totally edentulous arches, clinicians should consider designing two independent frames to facilitate the implant insertion path during the procedure (Figure 2). This issue is particularly important when high fixation zones are selected (EO);2.8.Specific surgical templates are recommended to guide the removal of the residual alveolar ridge (Figure 3). This will improve the adaptation of the CSI, facilitate its design, and reduce the risk of postoperative soft tissue dehiscence (EO);2.9.From a biomechanical perspective, there is no contraindication for connecting CSIs with previously placed conventional dental implants (EO);2.10.It is advisable to print a 3D model of the patient before surgery (EO).3.Surgical protocol and associated complications.
3.1.Although it is possible to place CSIs under local anesthesia, it is advisable to combine them with conscious sedation techniques or general anesthesia;3.2.Surgical asepsis guidelines must be followed during the procedure;3.3.The incision should be performed considering the final position of the keratinized mucosa since this tissue is essential to prevent long-term complications (EO);3.4.If the keratinized mucosa width is insufficient, it is advisable to perform soft tissue augmentation procedures;3.5.Soft tissue dehiscence leading to exposure of the CSI is the most common postoperative complication (Figure 4). This complication does not seem to affect CSI survival in the short term.3.6.Removal of the CSI is indicated when the implant has lost its stability or when recurrent infections occur;3.7.It is advisable to have a sterile 3D model of the patient present during the surgical procedure (Figure 5) (EO);3.8.After testing the insertion of the implant in the model, the CSI should be securely fixed with screws to the maxilla or mandible. The flap should be repositioned, leaving the abutments exposed.4.Prosthetic protocol and associated complications.
4.1.A thorough and complete preoperative prosthetic diagnosis is mandatory. This prosthetic planning is essential for designing the CSI correctly;4.2.The prosthodontic treatment principles and steps used in rehabilitation with conventional dental implants should be followed when using CSI. It is essential to create a prosthesis with ovoid pontics that allows correct assessment for oral hygiene;4.3.The clinical results of this group of experts support the use of fixed screw-retained restorations over CSIs. The literature also reports on the use of other types of rehabilitation (EO);4.4.The CSI can be immediately loaded;4.5.The provisional and definitive prostheses should not apply pressure on the soft tissues (EO);4.6.A minimum of 4 prosthetic connections are required to rehabilitate an entire arch;4.7.Whenever possible, the use of transepithelial abutments should be considered (EO);4.8.The materials employed in conventional implant-supported prostheses are also suitable for rehabilitation with CSIs;4.9.It is advisable to use an occlusal splint after the prosthetic rehabilitation to prevent the occurrence of mechanical complications, especially when the patient has natural dentition or a fixed implant-supported rehabilitation in the opposing arch (EO).5.Peri-implant supportive therapy.
5.1.There is no specific evidence reporting the maintenance protocol for CSI restorations;5.2.Control visits are recommended every 6 months to avoid or diagnose biological (e.g., bone loss under CSIs) or mechanical complications (e.g., prosthetic fracture) (EO);5.3.The main goal of peri-implant supportive therapy is to remove plaque accumulation and biofilm around implant abutments and prostheses. In the case of screw-retained restorations, these can be removed to thoroughly clean the surfaces (EO);5.4.Patients should be informed of the importance of these visits for the long-term maintenance of their rehabilitation and of the most common pathologies or complications. Patients should also be advised to seek clinical attention in cases of CSI mobility or soft tissue dehiscences (CSI exposure) (EO);5.5.These visits should include professional advice in case of risk factors/indicators. Patients should be informed that redness, bleeding, or inflammation of the peri-implant mucosa are important signs that, if left untreated, might result in significant long-term complications (EO).6.General recommendations and future perspectives (EO).
6.1.The available data on the use of CSIs are very scarce, precluding the establishment of clinical recommendations based on scientific evidence. It is essential to perform randomized clinical trials to compare the use of CSIs with other therapeutic alternatives. Additionally, cohort studies with a long-term follow-up could help determine the incidence, repercussions, and prognosis of complications associated with CSIs;6.2.Finite analysis studies evaluating different CSI designs would be desirable;6.3.Professionals are encouraged to undergo specific training in the use of CSIs;6.4.Professionals could benefit from developing additional tools or guides to reduce the margin of error. The creation of specifically designed custom guides for all steps of the treatment would be desirable;6.5.The development of specific prosthetic connections for CSIs might be interesting.

## 4. Discussion

The present review shows that the available data supporting the use of CSI are very scarce. Indeed, most reports are based on case series or retrospective cohort studies with very limited follow-up. In addition, several clinically relevant issues, like the repercussions of peri-implantitis on the long-term prognosis of these devices and which materials are the most suitable for the final prosthesis, are still unclear. Furthermore, because of technological advances, CSIs are constantly being improved, so it is likely that the reported data cannot be fully extrapolated to the present situation. For this reason, we believe that an expert consensus might provide valuable information to clinicians with limited experience in the use of CSIs to rehabilitate edentulous atrophic jaws.

Most authors [18,19,21] and the expert panel agree that these implants should be used when conventional implants cannot be placed or when complex bone regeneration techniques would be required. This patient profile is usually challenging to treat since resorption of the alveolar ridge might contraindicate fixed restorations and compromise the stability of a removable prosthesis. Indeed, when large vertical alveolar ridge defects are present, bone grafting techniques seem to have a higher incidence of complications. Alotaibi et al. [22] performed a network meta-analysis to compare the results associated with the use of onlay and inlay grafts, several types of membranes (resorbable and nonresorbable), distraction osteogenesis, tissue expansion, and short implants. These authors concluded that all grafting options (except the use of resorbable membranes) were associated with a statistically significant higher odds ratio of complications [22]. It is also important to stress that when extraoral bone harvesting is required, patients might experience pain in the donor site area and gait and sensory disturbances if the iliac crest is involved [23]. Furthermore, complex bone grafting procedures also limit the use of provisional prostheses since they might increase the risk of soft tissue dehiscences. Thus, CSIs seem to be a promising treatment option to provide fixed restorations to patients without the above-mentioned disadvantages. CSIs also allow faster recovery of the patient’s function and quality of life since these devices can be loaded immediately [12,17,19,20,24]. In general, CSIs might be used to support fixed full-arch prostheses or even partial-arch restorations [12,17,20]. Moreover, some authors have rehabilitated edentulous patients with CSI-retained overdentures with good outcomes [16].

According to several authors [25,26], zygomatic implants might also be an excellent alternative to bone grafting procedures in atrophic maxillas. Indeed, these implants seem to have excellent results even when immediate loading protocols are applied [25,26]. A recent systematic review has compared the outcomes of zygomatic implants placed with two different techniques (an original surgical technique and an anatomy-guided approach) and showed similar outcomes with survival rates higher than 90% for both options [27]. However, these authors also pointed out that sinusitis and soft tissue infection around the implant are common in these cases. Thus, CSIs might be a preferable option in patients with a previous history of maxillary sinus pathology.

As in any other treatment, a thorough preoperative diagnosis is paramount to achieve a successful outcome. In this regard, clinicians should obtain an in-depth medical history, perform a complete intra and extraoral examination, request high-quality computer tomography, and perform comprehensive prosthetic planning before the surgical procedure. The introduction of new technology, such as CSIs, should be carried out gradually, usually by professionals experienced in the field of implant dentistry. Indeed, there are no data about the learning curve needed to master this type of procedure. Moreover, some biological (soft tissue dehiscence or peri-implantitis) or mechanical complications (fractures) related to this treatment have been reported. Likewise, digital planning and the use of printed models may reduce fitting problems that can lead to the failure of these implants due to the mobility of the structure [18].

In this regard, it is important to stress that both surgical and prosthodontic factors must be considered to avoid complications. Thus, it is essential to design the CSIs, taking the final prostheses into account [28]. Equally, since CSIs are fully customized implants that must be perfectly adjusted to the patient’s anatomy, a CT scan of excellent quality is mandatory, although some authors use cone-beam computed tomography with adequate results [12,17,18,19,24]. The dataset should be checked to rule out defective slices in the anatomical region to be treated, e.g., caused by metal-induced scattering or motion artifacts [28]. During the surgical procedure, the surgeon must achieve a passive fit and perfect fixation of the implant since this is critical to avoid failure due to movements of the structure. Moreover, a 3D-printed model of the patient could be very useful for assessing the CSI adjustment preoperatively [12,17,18,19,20].

Most reports mention that intraoperative complications are uncommon. However, postoperative CSI exposure due to soft tissue dehiscence seems to be a frequent event. Thus, correct incision design and soft tissue grafting might reduce the incidence of this complication. This is an important issue since patients with atrophic jaws usually have an insufficient width of keratinized tissues, especially in the mandible [17]. The CSI design should also be adapted to prevent dehiscences. Sharp areas and abrupt transitions between the structure and the prosthetic connection areas should be avoided, and a polished surface might be preferable to avoid biofilm adhesion in case of exposure (Figure 1 and Figure 4) [28]. Fortunately, CSI exposure does not seem to compromise the short-term survival of the implant. Nemtoi et al. [19] reported several cases with CSI exposure that remained under function. However, this topic needs further research since this complication might have a long-term impact on the survival of the implants.

Information on the long-term prognosis of these restorations is scarce. Regarding biological complications, peri-implantitis is a common finding in conventional implants [9] and might also affect CSIs. Since peri-implantitis is associated with biofilm accumulation, patients should be included in peri-implant supportive therapy programs. Rams et al. [16] have identified anaerobic orange and red cluster bacteria in cobalt–chromium–molybdenum alloy CSIs. Although this material is not ideal and might increase the risk of infection and bone loss [9], a similar microbiota is likely to be found in both conventional and customized subperiosteal implants [16]. There are no studies giving specific information about the maintenance protocol for these restorations. Screw-retained restorations can be removed for professional hygiene, to remove plaque and biofilm from the prosthesis and CSIs, and to avoid soft tissue inflammation or infection through soft tissue dehiscence. In conventional dental implants, it has been observed that patients have little access to information about implant maintenance and peri-implant diseases. In fact, it has been shown that about half of the patients have not been informed about peri-implant diseases, and many of them have unrealistic information about the duration of this treatment, thinking that it is a lifelong treatment [29,30].

The lack of information provided to the patients could be linked to irregular maintenance visits, which, in turn, are related to increased pathology. Although CSIs are a distinct technology, they should undergo proper examination to evaluate all prosthetic components and check occlusion. Bone loss under CSIs could induce mobility of the structure. Consequently, it is of great importance to establish individualized maintenance intervals for each patient, usually every 5–6 months, according to risk indicators (e.g., periodontally compromised patients or patients with non-hygienic restorations), to remove bacterial plaque and biofilm and to assess peri-implant health status [31].

It is worth noting that one in five patients who do not attend a regular maintenance program may suffer from peri-implantitis at 5 years [32], and compliance with maintenance visits can reduce the occurrence of peri-implantitis by up to 25% [33]. During implant maintenance visits, special attention and professional advice should be given regarding the risk factors/indicators that have been associated with peri-implant disease, such as a history of periodontal disease or poor oral hygiene [34].

Regarding the risk of mechanical complications, the studies included in the present literature review have mainly reported some cases of fractured provisional prostheses. These events are also frequent in patients rehabilitated with conventional implants and are generally minor complications that can be solved without having to send the prosthesis to the laboratory for repair. Parafunctional habits such as bruxism and maxillary restorations seem to be variables linked to fractures of provisional prostheses. In definitive restorations, material chipping tends to appear at the follow-up. Review of occlusal contacts and checkup visits are necessary to avoid these complications or to diagnose them at an early stage. Fortunately, these minor complications do not seem to affect patients’ quality of life [35,36]. However, it might be advisable to use an occlusal splint after prosthetic rehabilitation, especially in patients with parafunctions. This could be placed even in the provisional period to ensure that rehabilitation is maintained throughout the interim period [36].

Rehabilitation of large edentulous sections improves the patients’ aesthetics and masticatory function. There are no specific success criteria for CSIs, and it is understood that the presence of soft tissue dehiscence could be a determining factor, facilitating the emergence of peri-implantitis. On the other hand, patient perceptions of treatment outcomes and quality of life are necessary variables to ascertain the success of the treatment [37], and the studies published so far do not provide these data. Patient-reported outcome measures were introduced at the 8th European Workshop on Periodontology [38] with the aim of improving the assessment of treatment outcomes according to patients’ perceptions and not only through clinical parameters. The use of psychometric tools validated for this context, such as the Oral Health Impact Profile (OHIP)-14 questionnaire or visual analog scales where the patient can objectify his or her satisfaction with the treatment at the level of aesthetics or mastication, should be systematically reported. In this way, the patient’s perception would be included in the criteria for measuring the success or outcome of a treatment. In fact, long-term reporting of repeated measures during the whole postoperative period and the prosthetic restoration could provide results regarding the maintenance of patients’ quality of life and the influence of any biological or mechanical complications.

This paper has important limitations that need to be discussed. Firstly, the number of available studies on CSIs is insufficient. Furthermore, most of these studies are retrospective, include a limited number of patients, have a short follow-up period, and present a high risk of bias. Secondly, the conclusions derived from the consensus meeting provide a low degree of recommendation. Finally, since no studies have been conducted to compare the use of CSIs with other treatment options, it cannot be asserted that CSI-supported restorations are the treatment of choice to rehabilitate patients with severely atrophic jaws. Therefore, randomized clinical trials (RCT) comparing the use of CSIs with zygomatic implants, with short or ultrashort implants, and with advanced bone regeneration procedures should be conducted in the future.

## 5. Conclusions

Customized subperiosteal implants (CSIs) are a promising treatment option to rehabilitate edentulous patients with atrophic jaws where conventional dental implants cannot be placed or as an alternative to complex regeneration procedures. These devices seem to have an excellent short-term survival rate, a low incidence of relevant complications, and less morbidity than complex bone grafting procedures. However, the scarcity of available data on the use of CSIs precludes the establishment of clinical recommendations based on scientific evidence.

## Figures and Tables

**Figure 1 biomimetics-09-00061-f001:**
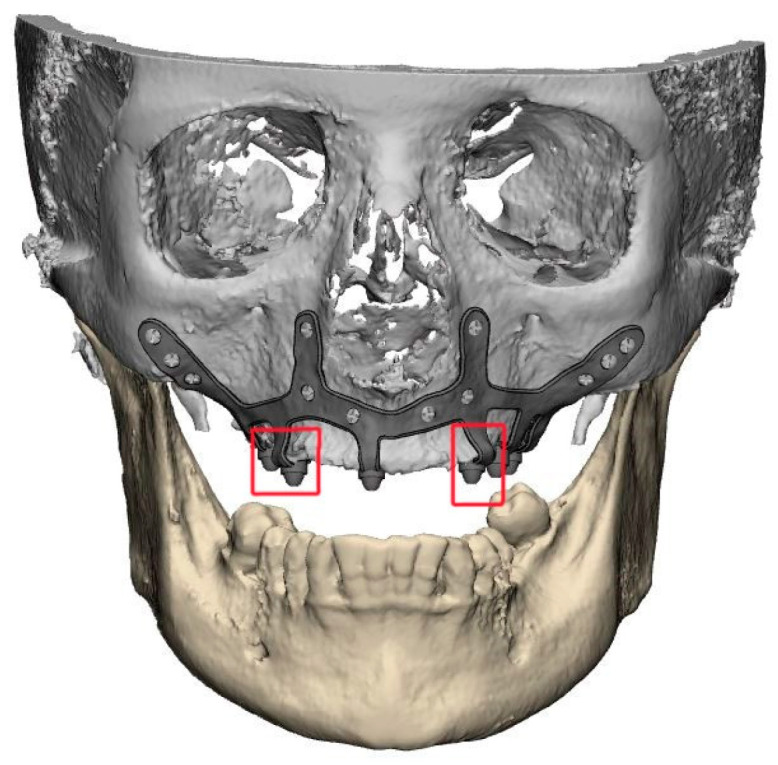
Example of a suitably designed customized subperiosteal implant (CSI) with smooth transitions (without sharp angles) between the frame and the prosthetic connections.

**Figure 2 biomimetics-09-00061-f002:**
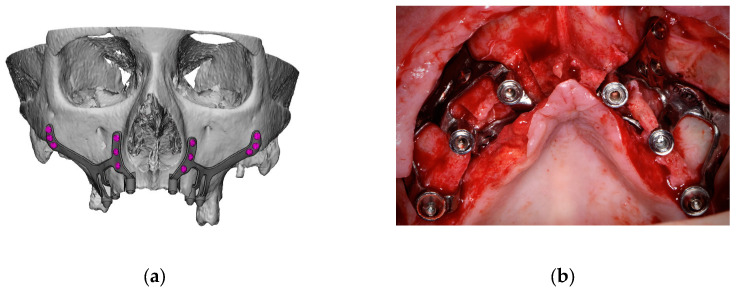
Example of a two-piece customized subperiosteal implant (CSI) designed for the patient’s specific anatomy. (**a**) Digital planning. (**b**) CSI placement.

**Figure 3 biomimetics-09-00061-f003:**
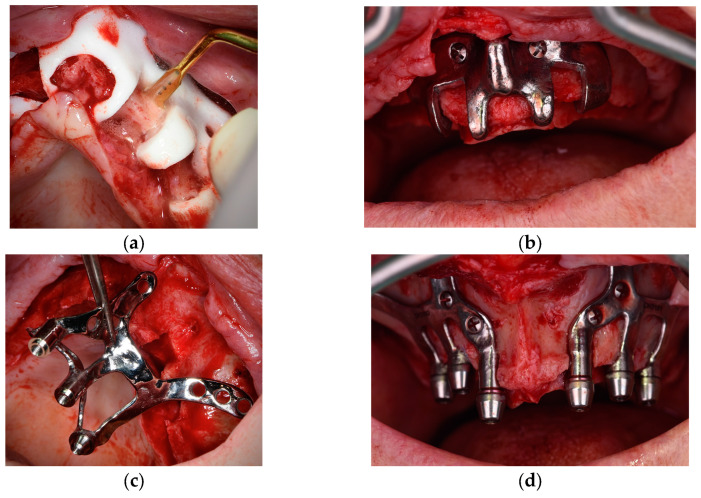
Surgical template used to remove the residual alveolar ridge. (**a**) Polyamide template. (**b**) Titanium alloy template. (**c**,**d**) CSI placement.

**Figure 4 biomimetics-09-00061-f004:**
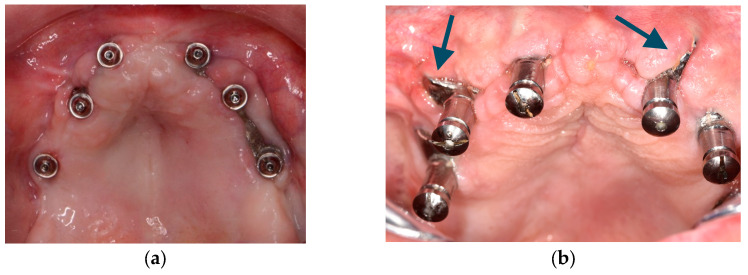
Clinical images of soft tissue dehiscences. These are some of the most common complications associated with customized subperiosteal implants (CSI) but do not seem to affect the short-term success rate of the treatment. (**a**) Exposure of a maxillary CSI caused by a soft tissue dehiscence. (**b**) Soft tissue dehiscence and peri-implant mucosa inflammation in a maxillary CSI probably related to the abrupt transition between the frame and the prosthetic connection (see arrows).

**Figure 5 biomimetics-09-00061-f005:**
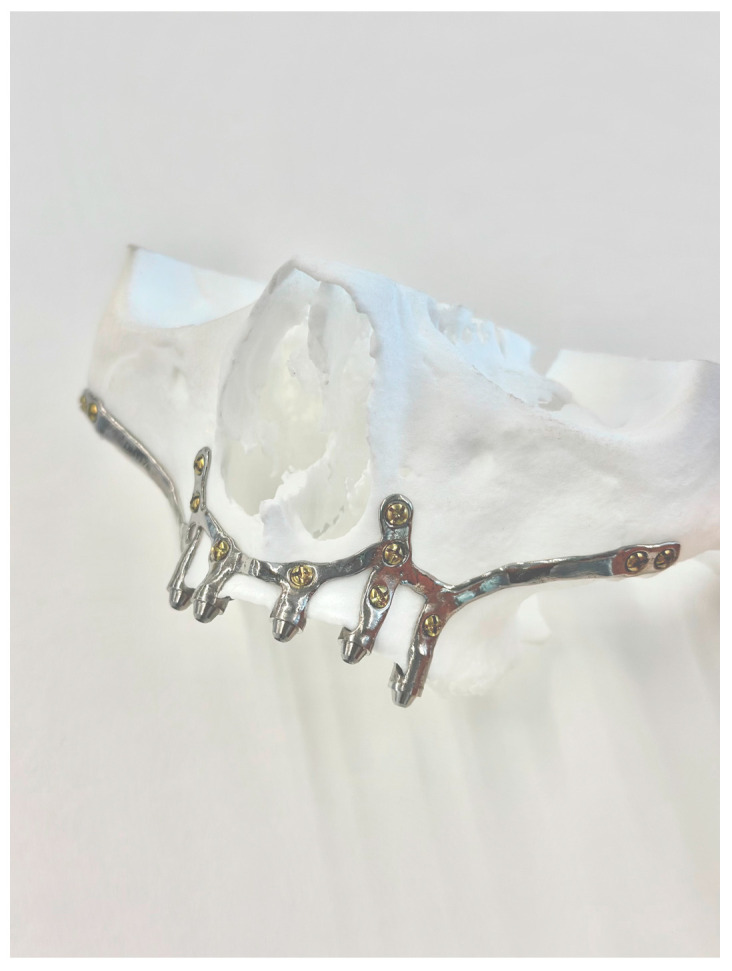
3D model of the patient with a customized subperiosteal implant (CSI).

**Table 1 biomimetics-09-00061-t001:** Main characteristics of the studies included in the review.

Author, Year, and Country	Study Design	Number of Patients	Indication	Type of Prosthesis	Type of Implant	Complications	Follow-Up	Level of Evidence (SIGN)
Rams et al., 2013 [15], United States of America	Case series	11	Edentulous mandible	Overdenture	Implant frames were cast using a cobalt–chromium–molybdenum alloy (Vitallium)	3 periimplantitis	11.7 years ± 1.5 years (range 10–13 years) in patients with periimplantitis; 2.4 years ± 4.9 years (range 9–22 years) in healthy patients	3
Cerea and Dolcini 2018 [12], Italy	Retrospective cohort	70	Total or partial edentulism	Provisional prosthesis (resin), Definitive cement-retained metal–ceramic prosthesis	Laser sintering titanium CSI	3 failures due to infections; 4 patients reported postoperative pain and swelling; 1 patient with recurrent infections 4 fractures of the provisional prosthesis; 2 patients with ceramic fractures (chipping) in the definitive prosthesis	2 years	2++
Mangano et al., 2020 [16], Russia	Case series	10	Partial posterior mandibular edentulism	Cement-retained provisional prosthesis (PMMA) 10 days after surgery. New provisional prosthesis 1 month after surgery. Definitive cement-retained prosthesis (zirconia–ceramic)	Laser sintering titanium CSI (titanium grade 5 micro-powders)	1 patient with postoperative pain and swelling; 2 patients with provisional prosthesis fractures	1 year	3
Cebrián-Carretero et al., 2022 [17], Spain	Case series	4	Oncological defects	Provisional prosthesis Fixed metal–ceramic prosthesis	Laser sintering titanium CSI	No complications	32 months (range 9 months–3 years)	3
Nemtoi et al., 2022 [18], Romania	Retrospective cohort	16	Edentulous maxilla (*n* = 10) Partially edentulous maxilla (*n* = 1) Edentulous mandible (*n* = 4) Partially edentulous mandible (*n* = 1)	Provisional prosthesis (resin) Fixed prosthesis (unspecified)	Laser sintering titanium CSI	1 failure due to incorrect adjustment and recurrent infections; 6 soft tissue dehiscences leading to CSI exposure 1 fracture of the provisional prosthesis;	6 months	2++
Dimitroulis et al., 2023 [19], Australia	Case series	21	Edentulous maxilla (*n* = 15) Edentulous mandible (*n* = 3) Partial edentulism (*n* = 2) Maxillectomy (*n* = 1)	Screw-retained provisional prosthesis (resin) Definitive prosthesis	Laser sintering titanium CSI	5 patients with CSI exposure: - 3 needed new CSI - 2 had a follow-up; 1 patient with implant mobility (additional retention screws were placed) 1 CSI was removed due to systemic causes (psychiatric disorder)	22.1 months (range 5–57 months)	3

CSI—Customized subperiosteal implant; *n*—number of patients; PMMA—polymethyl-methacrylate; SIGN—Scottish Intercollegiate Guidelines Network.

## Data Availability

No new data were created.

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
