# Peer review of "Customized Subperiosteal Implants for the Rehabilitation of Atrophic Jaws: A Consensus Report and Literature Review"

_biomimetics, 2024, doi:10.3390/biomimetics9010061_

Round 1
Reviewer 1 Report
Comments and Suggestions for Authors
This is a good paper,the customized subperiosteal implant is a treatment option to help for edentulous patients with atrophic jaws . The indications and contraindications of the customized subperiosteal imlants are well discussed.
The clinical guidelines of the surgery and the prosthetic rehabilitation is important.
It is true to say that the rehabilitation of the edentulous mandible or maxilla allows for the improvementof aesthetics and masticatory function in patients.
The manuscript is accept in present form.
Author Response
Thank you very much for your review and comments.
Reviewer 2 Report
Comments and Suggestions for Authors
Dear authors.
This paper deals with a new and innovative topic. As you rightly say, the literature is very scarce, with a small number of cases, a short follow-up, and a lack of important information: what happens when peri-implantitis affects a part of the structure, the importance of antagonists, final screw-retained or cemented prosthesis, problems with mandibular flexion, materials or techniques for the final prosthesis...
Therefore, from my point of view, it would increase the information of the expert consensus and would explain more precisely the correct steps of the surgical and prosthetic protocol.
For this reason, I think the title of the article should be different, focusing on the Consensus and not on the literature review.
Thank you for your article
Author Response
This paper deals with a new and innovative topic. As you rightly say, the literature is very scarce, with a small number of cases, a short follow-up, and a lack of important information: what happens when peri-implantitis affects a part of the structure, the importance of antagonists, final screw-retained or cemented prosthesis, problems with mandibular flexion, materials or techniques for the final prosthesis...
Therefore, from my point of view, it would increase the information of the expert consensus and would explain more precisely the correct steps of the surgical and prosthetic protocol.
Answer: Thank you for your suggestions. We have increased the information regarding the surgical (subheading 3.8) and prosthetic (subheadings 4.3 and 4.9) protocol.
For this reason, I think the title of the article should be different, focusing on the Consensus and not on the literature review.
Answer: We have changed the title to underline that this is mainly a consensus report paper.
Reviewer 3 Report
Comments and Suggestions for Authors
This is a narrative literature review and consensus report on a novel approach on customized subperiosteal implants for the rehabilitation of atrophic jaws. Although there were only a few articles, it is a timely publication to synthesize recent 3 articles and give a prospective view of it. Some comments as follows:
1. Current paragraphes do not have cohesive structure. Connecting sentences and logics should help to revise a apprehensive article.
2. The workflow to come up with the "consensus report" should be described and each statement ideally should link with the available literature or based on expert's opinion (And if there was any pro and against debate and discussion)
3. The complications rate should be calculated based on the included literature and included in the abstract. Otherwise, current description is very subjective and can be misleading
4. Photos of cases with soft tissue dehiscence around prosthetic abutments should be presented and discussed, since it was also mentioned by the authors that it is especially relevant. And in the protocol about explaining to the patients- how?
Need major revision to construct the paragraphes better and cohesive
Author Response
This is a narrative literature review and consensus report on a novel approach on customized subperiosteal implants for the rehabilitation of atrophic jaws. Although there were only a few articles, it is a timely publication to synthesize recent 3 articles and give a prospective view of it.
Answer: Thank you for your kind words.
Some comments as follows:
- Current paragraphes do not have cohesive structure. Connecting sentences and logics should help to revise a apprehensive article.
Answer: We have improved the structure of the paper. The text has also been reviewed by an experienced native English medical translator.
The workflow to come up with the "consensus report" should be described and each statement ideally should link with the available literature or based on expert's opinion (And if there was any pro and against debate and discussion).
Answer: We have expanded the material and methods section to explain the workflow of the consensus report. Concerning the available literature supporting most of the recommendations, the references have been added to subheading 3.2: Clinical guidelines based on the results of the consensus meeting. In addition, we have labeled the statements based solely on the experts’ opinion as “EO”.
- The complications rate should be calculated based on the included literature and included in the abstract. Otherwise, current description is very subjective and can be misleading.
Answer: The biological and mechanical complication rates are now presented in the abstract and results sections.
Photos of cases with soft tissue dehiscence around prosthetic abutments should be presented and discussed, since it was also mentioned by the authors that it is especially relevant. And in the protocol about explaining to the patients- how?
Answer: We have added a new figure (Figure 4b) with a case of soft tissue dehiscence. The figure legend has also been modified. Sections 5.4 and 5.5 of the consensus report have been changed to underline the importance of peri-implant supportive therapy in the prevention of complications.
We have also increased the length of the manuscript, as requested by the editor.
We sincerely hope that the changes made have significantly improved the manuscript.
Round 2
Reviewer 3 Report
Comments and Suggestions for Authors
Thank you for addressing the comments
Comments on the Quality of English LanguageOk